# Advancing Sleep Disorder Diagnostics: A Transformer-based EEG Model for Sleep Stage Classification and OSA Prediction

Cheng Wan[†], Micky C. Nnamdi[†], Wenqi Shi[†], Benjamin Smith [§], Chad Purnell[§], and May D. Wang[†]

[†] Georgia Institute of Technology, Atlanta, Georgia 30332–0250

[§] Shriners Children's, Chicago, USA

*Abstract*—**Sleep disorders, particularly Obstructive Sleep Apnea (OSA), have a considerable effect on an individual's health and quality of life. Accurate sleep stage classification and prediction of OSA are crucial for timely diagnosis and effective management of sleep disorders. In this study, we propose a sequential network that enhances sleep stage classification by incorporating self-attention mechanisms and Conditional Random Fields (CRF) into a deep learning model comprising multi-kernel Convolutional Neural Networks (CNNs) and Transformer-based encoders. The self-attention mechanism enables the model to focus on the most discriminative features extracted from single-channel electroencephalography (EEG) recordings, while the CRF module captures the temporal dependencies between sleep stages, improving the model's ability to learn more plausible sleep stage sequences. Moreover, we explore the relationship between sleep stages and OSA severity by utilizing the predicted sleep stage features to train various regression models for Apnea-Hypopnea Index (AHI) prediction. Our experiments demonstrate an improved sleep stage classification performance of 78.7%, particularly on datasets with diverse AHI values, and highlight the potential of leveraging sleep stage information for monitoring OSA. By employing advanced deep learning techniques, we thoroughly explore the intricate relationship between sleep stages and sleep apnea, laying the foundation for more precise and automated diagnostics of sleep disorders.**

*Index Terms*—**sleep stage classification, obstructive sleep apnea, transformer, clinical decision support.**

## I. INTRODUCTION

Sleep is a crucial aspect of human health and well-being, with disorders such as sleep apnea significantly impacting an individual's quality of life [1], [2]. Obstructive Sleep Apnea (OSA) is a common sleep disorder characterized by repeated episodes of partial or complete upper airway obstruction during sleep [3], leading to intermittent hypoxia and sleep fragmentation [4], [5]. The disruption of normal sleeping behavior in patients with OSA is associated with severe comorbid diseases such as hypertension, diabetes, cardiovascular disease, and stroke, making early detection and treatment crucial [6]. The severity of OSA is quantified using the Apnea-Hypopnea Index (AHI), which measures the number of apnea and hypopnea events per hour of sleep. OSA severity is categorized

This research has been supported by a seed research grant from Shriners Children's, a Wallace H. Coulter Distinguished Faculty Fellowship, a Petit Institute Faculty Fellowship, and research funding from Amazon and Microsoft Research to Professor May D. Wang. Correspondence to: May D. Wang (maywang@gatech.edu).

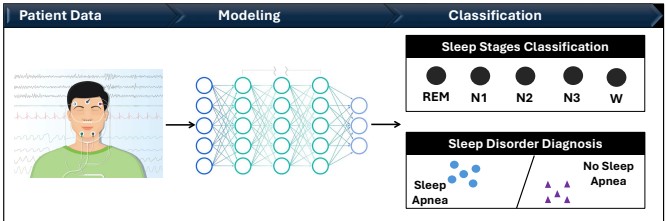

Fig. 1: An overview of the proposed sequential network. We first collect the whole record of single-channel EEG signals, sleep stage information, and AHI labels from patients, then model for both sleep stages and sleep apnea with EEG signals.

into four levels: normal (AHI $< 5$), mild ($5 \le$ AHI $< 15$), moderate ($15 \le$ AHI $< 30$), and severe (AHI $\ge 30$) [7], [8]. Sleep stages, however, are distinct phases of sleep characterized by specific brain activity patterns, eye movements, and muscle tone [9], [10]. These stages include wakefulness (W), rapid eye movement (REM), and four Non-REM stages (N1, N2, N3, and N4), ranging from light to deep sleep [11]. The distribution and cycling of these stages throughout the night provide valuable insights into an individual's sleep quality and health.

Breakthroughs in computational intelligence and neural architectures have unlocked innovative methods for interpreting complex neurophysiological data, especially electroencephalography (EEG) recordings [12]–[16]. Advanced deep learning methods, such as Convolutional Neural Networks (CNNs) and Transformer architectures, have demonstrated notable success in tasks classifying sleep stages and detecting sleep disorders. These models are proficient in capturing time-based dependencies and intricate patterns within sequential datasets, making them highly effective for analyzing physiological signals. Consequently, these advancements are facilitating the development of more automated and reliable systems in clinical practice.

EEG signals have been widely used for diagnosing various sleep-related issues, including sleep staging and the detection of sleep disorders [17]. These EEG-based diagnostic models have proven effective due to their ability to capture the brain's electrical activity and analyze nuances between sleep stages and abnormal pathology. Despite these advancements, most existing models separately focus on either sleep stage classi-

fication [13], [15], [18], [19] or sleep apnea detection [17], [20]–[22]. This separation is a limitation, as few studies have attempted to simultaneously classify sleep stages and the severity of sleep apnea using physiological signals. Early research suggests a link between sleep stages and the severity of sleep apnea, indicating that disruptions in sleep stages could be a sign of OSA [23], [24]. Understanding and leveraging this relationship is vital for clinical decision-making, providing comprehensive insights into a patient's sleep health and facilitating more accurate and holistic diagnoses. The capability to simultaneously classify sleep stages and predict sleep apnea severity using a single model can greatly improve the efficiency and efficacy of sleep disorder diagnostics.

Transformer-based models have recently become a powerful approach for modeling sequential physiological signals, as they can more effectively grasp distant dependencies and contextual nuances than traditional RNNs and LSTMs. [25], [26]. Transformers employ self-attention mechanisms to assign varying importance to different components of the input sequence, making them particularly suited for tasks where capturing complex temporal patterns is essential. In this work, we enhance the Transformer-based model with Conditional Random Fields (CRF) to improve the classification of sequential data by modeling the transition probabilities between different states, a crucial factor for precise classification in sleep stages. The CRF layer helps in capturing the temporal dependencies between sleep stages, ensuring more plausible and accurate sequence predictions. This combination leverages the strengths of both Transformer and CRF models, providing a powerful framework for sequential data analysis. The main contributions of this paper include the following:

- **Novel Sequential Network**: We introduce a novel architecture that simultaneously performs sleep stage classification and sleep apnea severity prediction using multi-kernel CNNs, self-attention mechanisms, and CRFs to capture temporal dependencies.
- **Dual Timescale Task Performance**: The model effectively handles short-term sleep stage classification and long-term sleep apnea severity prediction within a single framework by integrating self-attention and CRF layers.
- **Model Interpretability**: We employ attention heatmaps to shed light on the model's decision-making mechanisms, thereby improving the transparency and reliability of the predictions.

In summary, our work illustrates how integrating advanced neural network techniques with traditional probabilistic models can lead to more accurate and interpretable outcomes in sleep stage classification and sleep apnea severity prediction, ultimately contributing to better clinical decision support systems. By combining the strengths of Transformer-based models and CRFs, we develop a comprehensive framework for analyzing EEG signals and enhancing sleep disorder diagnostics.

## II. RELATED WORKS

### A. Sleep Stages Classification

Deep learning has been employed in different areas and has shown its superiority over conventional machine learning models without the need for domain knowledge. For instance, Tsinalis et al. addressed this classification task by utilizing a combination of convolutional and pooling layers, along with fully connected layers [27]. In a different study, Sors et al. utilized a model composed of 12 convolutional layers paired with two fully connected layers [18]. Additionally, Chambon et al. [28] introduced a two-dimensional convolution network integrated with MaxPooling layers to categorize the raw data obtained from three channels: EEG, EOG, and EMG. Sokolovsky et al. [29] developed a more complex CNN architecture, demonstrating that increasing the network's depth resulted in improved performance. Phan et al. [19] transformed raw signals into log-power spectra and employed a CNN to carry out both classification tasks aimed at recognizing sleep stages. While these CNN models have shown good performance in classifying sleep stages, they struggle to accurately capture the temporal relationships within the EEG data.

### B. Sleep Apnea Severity Classification

In the context of sleep apnea detection, Wang et al. [12] investigate the use of CNN and LSTM models to predict apnea events using respiratory data, presenting four advanced methods—1D-CNN, 1D-CNN-LSTM, ConvLSTM, and 2D-CNN-LSTM. Tested on a large dataset, these models demonstrated robust performance, achieving up to 83% sensitivity and 85% specificity, suggesting their potential to enhance apnea management. Sheta et al. [16] employs various classifiers, including a novel CNN-LSTM hybrid, to diagnose OSA using ECG data. The results highlighted the effectiveness of KNN and ensemble decision trees, with enhancements observed through ADASYN and feature selection techniques. The proposed CNN-LSTM model demonstrated superior diagnostic performance, suggesting potential methods for future research in feature selection and fuzzy logic to further enhance OSA detection.

Even though these studies have made important advancements in the field, there is still a need for research that examines the correlation between sleep stages and sleep apnea using deep learning models. Our study aims to bridge this gap by proposing a novel approach that combines sleep stage classification and OSA severity prediction. This will provide a more thorough comprehension of the relationship between these two crucial elements of sleep health.

## III. METHODS

### A. The Overall Sequential Network

The comprehensive designed architecture of our proposed model includes three primary parts: a Feature Extractor consists of feature extraction and self-attention enhancement; an Encoding Module consists of a transformer-based model, which is usually recognized as a temporal context encoding

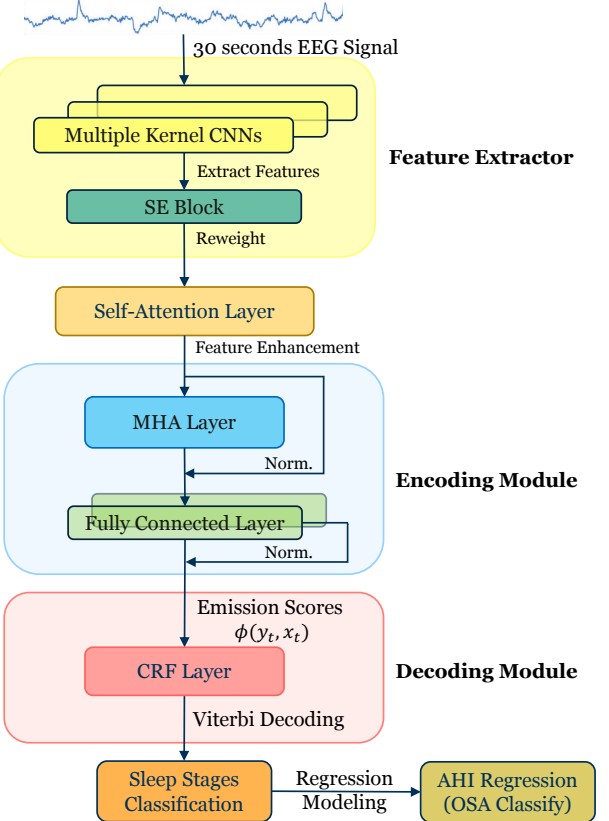

Fig. 2: Our proposed sequential network for classifying sleep stages and predicting sleep apnea via a transformer-based model, which includes three main components: feature extractor, encoding module, and decoding module. We utilize the features from predicted sleep stages to further classify normal and OSA subjects.

process; and a Decoder Module consists of a conditional random field for final classification. In detail, the EEG signal is first processed through a multi-kernel CNN to extract features, which are then recalibrated by the Squeeze-and-Excitation block (SE-Block). These recalibrated features are enhanced by an attention mechanism designed to obtain long-range dependencies. The enhanced features are then passed through a Transformer-based encoder to further capture temporal relationships. Finally, we incorporate a CRF layer to model the progression between different sleep stages, enabling the prediction of the most likely sequence of sleep stages.

For sleep apnea prediction, the output from the temporal encoder is passed through a sequence of dense layers to perform sleep stage classification. The model predicts a sleep stage to every 30 seconds segment. The sleep stage predictions for all segments from one record are then aggregated to obtain a representation of the entire EEG recording. This aggregated representation serves as input to a regression model, such as a dense neural network, for estimating the AHI value and determining the severity of OSA. Algorithm 1 presents the procedure for sleep stage classification and sleep apnea severity assessment.

## B. Feature Extractor

For EEG signal feature extraction, we utilize the multi-kernel CNN and feature recalibration module to form the feature extraction block, which has been verified to be an effective EEG feature extraction method [14], [15]. The multi-kernel CNN, as shown in Figure 2, processes the raw EEG signals through multiple convolutional layers to capture features at various scales. Following every convolutional layer, we implement batch normalization and a ReLU activation function. The convolutional window captures about 50 samples (0.5 seconds) of the EEG signal, enabling the model to learn local temporal dependencies. To prevent overfitting, we add a dropout layer with a rate of 0.5 after each max pooling operation.

The SE-Block aims to re-weight each channel of the features learned by the multi-kernel CNN. It is constructed from an adaptive average pooling layer with two dense layers that utilize a sigmoid activation function. The SE-Block learns to assign importance weights to different channels, enhancing the model's ability to identify and utilize the most informative features. The output of the SE-Block is then multiplied element-wise with the input features to obtain the recalibrated features.

Formally, let $\mathbf{X} \in \mathbb{R}^{B \times T \times C}$ be the input EEG recording, where $B$ corresponds to the batch size and $T$ denotes the temporal length, while $C$ represents the number of channels. Thus, the feature extraction is defined as:

$$\mathbf{f}_t = \sigma(\mathbf{W}_1 * \mathbf{x}_t + \mathbf{b}_1), \tag{1}$$

where $\mathbf{W}1$ and $\mathbf{b}1$ represent the convolutional layer's weights and biases, respectively, and $\sigma$ signifies the ReLU activation function. The SE-Block then recalibrates the features:

$$\tilde{\mathbf{f}}_t = \text{SE}(\mathbf{f}_t), \tag{2}$$

where SE denotes the Squeeze-and-Excitation block.

## C. Self-Attention in Feature Enhancement

To improve the model's capacity to concentrate on the most discriminative parts of the extracted features [30], we introduce a self-attention mechanism between the feature extraction and temporal context encoding stages. The self-attention mechanism captures long-range dependencies within the extracted features, which is crucial for accurate sleep stage classification.

The self-attention mechanism operates on the feature map $\mathbf{F} \in \mathbb{R}^{B \times T \times C}$. The self-attention is computed:

$$\text{Attention}(\mathbf{Q}, \mathbf{K}, \mathbf{V}) = \text{softmax}\left(\frac{\mathbf{Q}\mathbf{K}^\top}{\sqrt{d_k}}\right)\mathbf{V}, \tag{3}$$

where $\{\mathbf{Q}, \mathbf{K}, \mathbf{V}\}$ corresponds to the query, key, and value matrices, respectively, and $d_k$ represents the dimension of the key vectors. The feature map $\mathbf{F}$ undergoes an initial linear transformation to obtain $\{\mathbf{Q}, \mathbf{K}, \mathbf{V}\}$:

$$\mathbf{Q} = \mathbf{F}\mathbf{W}_Q, \quad \mathbf{K} = \mathbf{F}\mathbf{W}_K, \quad \mathbf{V} = \mathbf{F}\mathbf{W}_V, \tag{4}$$

where $\mathbf{W}_Q$, $\mathbf{W}_K$, and $\mathbf{W}_V$ represent the trainable weight matrices. Subsequently, the self-attention output $\mathbf{A}$ is calcu-

**Algorithm 1** Our Sequential Network for Sleep Stage & OSA Classification

1: **Input:** EEG recording $\mathbf{X} = [\mathbf{x}_1, \mathbf{x}_2, \ldots, \mathbf{x}_T]$, $T$ denotes the count of epochs
2: **Output:** Pred. sleep stage $\hat{\mathbf{Y}} = [\hat{y}_1, \hat{y}_2, \ldots, \hat{y}_T]$, Pred. AHI $\hat{a}$
3: **for** $t = 1$ to $T$ **do**
4:     $\mathbf{f}_t = \sigma(\mathbf{W}_1 * \mathbf{x}_t + \mathbf{b}_1)$        *// multi-kernel CNN*
5:     $\tilde{\mathbf{f}}_t = \text{SE}(\mathbf{f}_t)$                 *// SE-Block*
6: **end for**
7: $\tilde{\mathbf{F}} = [\tilde{\mathbf{f}}_1, \tilde{\mathbf{f}}_2, \ldots, \tilde{\mathbf{f}}_T]$
8: $\mathbf{Q}, \mathbf{K}, \mathbf{V} = \mathbf{W}_Q \tilde{\mathbf{F}}, \mathbf{W}_K \tilde{\mathbf{F}}, \mathbf{W}_V \tilde{\mathbf{F}}$
9: $\mathbf{A} = \text{softmax}\left(\frac{\mathbf{Q}\mathbf{K}^\top}{\sqrt{d_k}}\right)\mathbf{V}$        *// Self-Attention*
10: $\mathbf{E} = \text{LayerNorm}(\tilde{\mathbf{F}} + \mathbf{A})$
11: $\mathbf{H} = \mathbf{E}$
12: **for** $i = 1$ to $N$ **do**
13:     $\mathbf{H} = \text{LayerNorm}(\mathbf{H} + \text{MHA}(\mathbf{H}))$
                           *// Multi-Head Attention*
14:     $\mathbf{H} = \text{LayerNorm}(\mathbf{H} + \text{FFN}(\mathbf{H}))$
                           *// Feed-Forward Network*
15: **end for**
16: $\mathbf{Z} = \text{Flatten}(\mathbf{H})$
17: $\mathbf{U} = \mathbf{W}_{\text{fc}}\mathbf{Z} + \mathbf{b}_{\text{fc}}$        *// Emission Scores*
18: $\mathbf{U} = \mathbf{U}.view(T, -1)$        *// Reshape for CRF*
19: **CRF Decoding:**
20: Initialize $\alpha_1(y) = \exp(U_{1,y})$ for all $y$
21: **for** $t = 2$ to $T$ **do**
22:     **for** each state $y_t$ **do**
23:         $\alpha_t(y_t) = \sum_{y_{t-1}} \alpha_{t-1}(y_{t-1}) \cdot \exp(U_{t,y_t} + \text{trans}(y_{t-1}, y_t))$
24:     **end for**
25: **end for**
26: $\hat{y} = \arg\max_y \alpha_T(y)$        *// Viterbi Decoding*
27: $z = F(\hat{y})$
28: $\hat{a} = \text{Regression}(z)$

---

lated and added to the original feature map, followed by layer normalization:

$$\mathbf{A} = \text{Attention}(\mathbf{Q}, \mathbf{K}, \mathbf{V}), \qquad (5)$$

$$\mathbf{F}_{\text{attn}} = \text{LayerNorm}(\mathbf{F} + \mathbf{A}), \qquad (6)$$

### D. Transformer-based Encoder Module

The enhanced features from the self-attention block are passed through a Transformer-based module to capture long-range dependencies as a temporal context encoder in the EEG signal, which is adopted from [15], [31]. This encoder architecture utilizes the attention mechanism to model the temporal relationships between different segments of the entire recorded EEG signal. Formally, the Transformer-based encoder processes the input features $\mathbf{F}_{\text{attn}}$ through several layers composed of multiple attention heads and feed-forward neural networks with layer normalization:

$$\mathbf{H} = \mathbf{F}_{\text{attn}}, \qquad (7)$$

$$\mathbf{H} = \text{LayerNorm}(\mathbf{H} + \text{MHA}(\mathbf{H})), \qquad (8)$$

$$\mathbf{H} = \text{LayerNorm}(\mathbf{H} + \text{FFN}(\mathbf{H})), \qquad (9)$$

where MHA refers to the multi-head attention mechanism, and FFN stands for the feed-forward network.

### E. Conditional Random Field

Inspired by Fonseca et al. [32], we propose a classifier for enhancing the previous model by incorporating a CRF layer, which models the temporal relationships between sleep stages. In the context of sleep stage classification, CRF can explicitly capture the transition dynamics between sleep stages, helping the model learn more plausible sleep stage sequences. This is especially pertinent to our task, as sleep stages exhibit strong temporal patterns and dependencies, which may be affected by the occurrence and severity of sleep apnea. For example, patients with severe sleep apnea may experience more frequent transitions between sleep stages or have altered durations of specific sleep stages compared to those with mild or no sleep apnea. By leveraging CRF, we aim to capture these temporal dynamics and improve the model's ability to accurately classify sleep stages across a diverse range of sleep apnea severities.

To incorporate CRF into the proposed model, we add a CRF layer following the model's final output layer. The CRF layer models the transition probabilities between sleep stages, denoted as $\psi(y_{t-1}, y_t)$, representing the likelihood of transitioning from sleep stage $y_{t-1}$ to $y_t$. The emission scores, denoted as $\phi(y_t, x_t)$, are obtained from the proposed model's output and indicate the likelihood of observing a particular sleep stage $y_t$ at each time step $t$ given the input features $x_t$. The joint probability of a sequence of sleep stages $\mathbf{y} = (y_1, y_2, \ldots, y_T)$ given the input sequence $\mathbf{x} = (x_1, x_2, \ldots, x_T)$ is defined as:

$$P(\mathbf{y}|\mathbf{x}) = \frac{1}{Z(\mathbf{x})} \exp\left(\sum_{t=1}^{T} \phi(y_t, x_t) + \sum_{t=1}^{T} \psi(y_{t-1}, y_t)\right), \tag{10}$$

where $Z(\mathbf{x})$ is the normalization factor, also known as the partition function, which ensures that the probabilities sum to one:

$$Z(\mathbf{x}) = \sum_{\mathbf{y}} \exp\left(\sum_{t=1}^{T} \phi(y_t, x_t) + \sum_{t=1}^{T} \psi(y_{t-1}, y_t)\right). \tag{11}$$

To train this CRF-based model, we design a weighted CRF loss function that combines the negative log-likelihood loss of the CRF and a weighted cross-entropy loss. The negative log-likelihood loss encourages the model to learn the correct sequence of sleep stages, the weighted cross-entropy loss is employed to mitigate the issue of class imbalance in the distribution of sleep stages. The weighted CRF loss function is defined as:

$$\mathcal{L}_{CRF} = -\log P(\mathbf{y}|\mathbf{x}) + \sum_{t=1}^{T} \sum_{c=1}^{C} w_c \cdot y_{t,c} \log \hat{y}_{t,c}, \tag{12}$$

where $w_c$ is the weight for class $c$, $y_{t,c}$ is the true label for class $c$ at time step $t$, and $\hat{y}_{t,c}$ is the predicted probability for class $c$ at time step $t$. The first component of the loss function is the negative log-likelihood of the CRF, which encourages

TABLE I: Details of two different AHI distributed datasets containing 329 subjects from SHHS-1 study. SHHS 1-L represents subjects with AHI < 5, while SHHS 1-D includes subjects with diverse AHI values, reflecting varying severities of OSA.

| Dataset | AHI Category | Sampling Rate | Channel | REM | N1 | N2 | N3 | W | Total Samples |
|---|---|---|---|---|---|---|---|---|---|
| SHHS 1-L | AHI<5 | 125 Hz | C4-A1 | 65953 | 10304 | 142125 | 60153 | 46319 | 324854 |
| SHHS 1-D | Diverse AHI | 125 Hz | C4-A1 | 51244 | 11135 | 140826 | 47635 | 58096 | 308936 |

TABLE II: Comparison between our proposed model and cutting-edge models on two different AHI distributed datasets. The best performance is highlighted in bold.

| Method | Dataset | W | N1 | N2 | N3 | REM | Accuracy |
|---|---|---|---|---|---|---|---|
| AttnSleep [15] | SHHS 1-L | 84.7 | 45.7 | 80.9 | 76.7 | 77.9 | 79.2 |
| **Ours** | SHHS 1-L | 88.7 | 19.2 | 82.5 | **78.5** | 85.5 | **81.6** |
| w/o Self-Attn | SHHS 1-L | **90.0** | 18.1 | 82.6 | 75.5 | **85.8** | 81.5 |
| w/o CRF | SHHS 1-L | 86.6 | **58.5** | **86.9** | 64.4 | 65.1 | 78.0 |
| AttnSleep [15] | SHHS 1-D | 87.6 | **30.0** | 76.7 | 64.6 | 76.2 | 75.3 |
| **Ours** | SHHS 1-D | **88.6** | 6.2 | **80.8** | **75.1** | 76.8 | **78.7** |
| w/o Self-Attn | SHHS 1-D | 88.3 | 7.8 | 76.6 | 71.1 | 79.7 | 76.5 |
| w/o CRF | SHHS 1-D | 83.1 | 22.0 | 77.3 | 56.4 | **85.1** | 74.4 |

the model to correctly learn the sequence of sleep stages. The second component is the weighted cross-entropy loss, which helps address the class imbalance in sleep stage distribution.

During inference, the most likely sequence of sleep stages $\mathbf{y}^*$ is obtained using the Viterbi algorithm:

$$\mathbf{y}^* = \arg\max_{\mathbf{y}} P(\mathbf{y}|\mathbf{x}). \tag{13}$$

The Viterbi algorithm efficiently finds the most likely sequence by maximizing the conditional probability $P(\mathbf{y}|\mathbf{x})$, considering both the emission scores and transition scores. By incorporating the CRF layer and the weighted CRF loss function, our CRF-based model can better capture the temporal dependencies between sleep stages and improve its performance across a range of sleep apnea severities.

## IV. RESULTS AND DISCUSSION

### A. Dataset

#### 1) SHHS-1 Dataset

The Sleep Heart Health Study (SHHS) [33], [34] is an extensive multi-center cohort investigation that explores the wider impact of sleep-disordered breathing on cardiovascular well-being. The study includes participants with various health conditions, such as lung, cardiovascular, and coronary diseases. Contrary to prior studies, we did not solely select subjects with low AHI. Instead, we curated two distinct datasets, each containing 329 subjects:

1) **SHHS 1-L:** This group includes subjects with an AHI less than 5, representing individuals without significant sleep apnea, as per previous studies [32]. This dataset was chosen due to its stable sleep patterns, minimizing the interference of sleep-disordered breathing.

2) **SHHS 1-D:** This group consists of a random selection of subjects with no restrictions on AHI values. This dataset was assembled to test how well the model performs across a wide range of AHI values, similar to real-world conditions. By including different AHI levels, our goal is to evaluate how well the model works for a larger population and how adaptable it is. The diverse range of

AHI values aligns with the guidelines established by the American Academy of Sleep Medicine (AASM) [35]–[38], which categorize sleep apnea severity into four categories: no apnea (AHI < 5), mild apnea (5–15), moderate apnea (15–30), and severe apnea (AHI > 30).

In both datasets, EEG data from the C4-A1 channel was utilized, and sampled at a frequency of 125 Hz to maintain consistency across the analysis. These datasets enable a comparison of model performance across populations with different severities of sleep apnea, and help evaluate the generalizability of our approach.

#### 2) Data Preprocessing

EEG signals inherently contain uncertainties and imprecisions, which can significantly impact analysis results, making proper preprocessing crucial to improve signal quality and reduce artifacts [39]. Both datasets consist of single-channel EEG signals. For each subject's recording, we also documented the corresponding AHI value to facilitate the subsequent analysis of sleep apnea severity. The preprocessing steps applied to both datasets are as follows:

- Omission of any unidentified stages that do not correspond to the predefined sleep stages.
- Combining N3 and N4 stages into a unified stage (N3), as per AASM guidelines.
- Restricting wake periods to 30 minutes before and after sleep phases to emphasize the sleep stages.

### B. Sleep Stages Classification

#### 1) Experimental Settings

In alignment with the prior study by Eldele et al. (2021) [15], we utilize various models for training and testing on the SHHS-1 dataset, which includes two distinct AHI distributions. These models are employed to classify sleep stages using a 30-second EEG signal recorded from a single channel, in the same settings. The experimental configuration includes a batch size of 128, the use of the Adam optimizer with a learning rate of 1e-3, and a weight decay of 1e-3. The training process spans 100 epochs. Maintaining consistent settings across each experiment, as illustrated in Table II, our model achieves an overall classification accuracy of 81.6% for sleep stages on the first dataset SHHS 1-L, which is an improvement when compared to the previous cutting-edge model [15]. We also conduct ablation experiments by removing the self-attention module and the CRF module separately for comparison, thereby demonstrating the effectiveness of these two modules.

#### 2) Results and Analysis

As shown in the results of Table II, all models perform worst in predicting the N1 stage, with accuracies below 60%, suggesting that N1 is the most challenging of the five stages

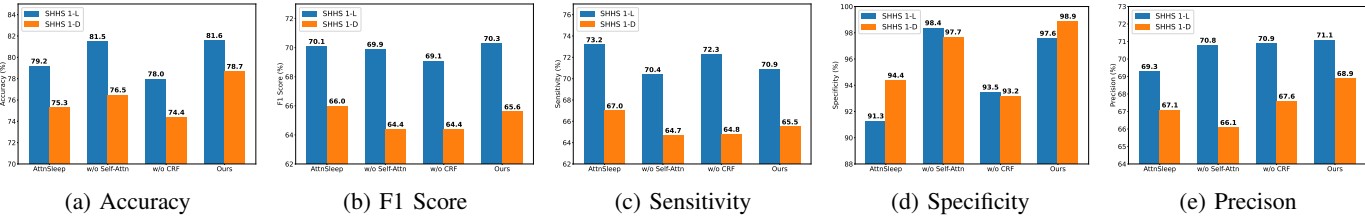

Fig. 3: Comparison of testing results for four models across five metrics (Accuracy, F1 Score, Sensitivity, Specificity, and Precision) on two datasets.

for models to capture and predict. The weighted CRF greatly enhances the model's performance in comparison to using the previous model alone, particularly on the dataset with diverse AHI values. Regarding the prediction results on the two different datasets, we find that the different AHI distributions of the datasets greatly influence the model's performance. Modeling on the dataset with diverse AHI shows a significant decrease in prediction accuracy about 10.1% compared to the dataset with AHI less than 5 according to our method. This suggests that previous studies using datasets composed of normal subjects for modeling may obtain relatively stable EEG signals and changes of sleep stages to achieve better prediction results. These results support trends explored by Shahveisi et. al. identifying temporal differences between sleep stages of patients with OSA and healthy individuals, specifically regarding the duration of N1 stages [6]. With the increased variability in temporal trends due to the inclusion of OSA patients in the dataset with diverse AHI values representing different severities, the reduced prediction accuracy further highlights the interaction between sleep stages and sleep apnea.

### 3) Explainability of Model Decision Making

We utilize the attention mechanism from our trained model to analyze EEG signals across different sleep stages, visualizing neural features through importance heatmaps as Figure 4. The heatmaps (a-e) reveal the model's focus on various segments of the EEG signals, reflecting the distinct neural activities characteristic of each sleep stage. In the Wake stage (4a), the model primarily focuses on two distinct areas with noticeable fluctuations, alongside a stable segment, suggesting attention to changes in alertness and restfulness. During the N1 stage (4b), the model attends to multiple fluctuation areas, particularly emphasizing two larger oscillations, indicative of the shift from wakefulness to sleep. In N2 (4c), the model concentrates on the region with the most pronounced fluctuation, covering both peaks and troughs, which may align with the presence of sleep spindles or K-complexes—key markers of this stage. In the N3 stage (4d), the model highlights two regions of significant fluctuation, reflecting the slow-wave activity that defines deep sleep. Finally, in the REM stage (4e), attention is directed towards a combination of peak areas and a stable region, likely correlating with vivid dreaming and high brain activity.

### C. Sleep Apnea Classification

After obtaining the sleep stage classification results from the previous stage, we extract different features from the classification results to form dataset with AHI labels. The training and testing datasets are simply divided in a 7:3 ratio to be further input into the regression model to learn AHI, thereby exploring the possibility of directly predicting OSA from sleep stages. Since each subject's record consists of many sleep stage segments corresponding to one OSA label, we perform simple feature extraction on the predicted sleep stages from the previous step, such as the average of predicted sleep stages and the proportion of each kind of stage, and use them as features to input into various traditional regression models (Random Forest, Linear Regressor, Logistic Regressor). Ultimately, we obtain the OSA classification results by AHI predictions from different models with various features. OSA is classified based on the corresponding predicted AHI value.

As shown in Figure 6, we obtain the testing results for both accuracy and F1-score regarding the 4-class classification related to OSA using predicted sleep stages. It can be observed that the results using the distribution across the five distinct sleep stages as features in the logistic regressor achieve the highest accuracy of 54.55%. It is worth noting that the overall framework has only a 71.07% testing result on sleep stage prediction in the last step, while the accuracy of the four-class classification for OSA reaches 54.55%. After we further perform binary classification of OSA (normal or OSA) as most studies have done [12], [40], we find that the testing accuracy reaches 67.68%, with the per-class F1-score results being 0.76 for OSA and 0.50 for normal in the test set. Furthermore, based on the box plot analysis 5, several key differences in sleep stage proportions between OSA patients and normal individuals are observed. OSA patients exhibit a higher proportion in the Wake stage (p-value = 1.756e-13), which likely results from frequent awakenings and light sleep caused by the reduced blood oxygen saturation indicative of sleep apnea. In contrast, normal individuals have a higher proportion of deep sleep (N3, p-value = 2.233e-07), indicating their ability to maintain a healthy sleep stage cycle. Both groups show similar distributions in the N2 stage (p-value = 7.986e-01), suggesting minimal differences in this sleep stage between OSA patients and normal individuals. The differences observed in N1 (p-value = 6.194e-08) and REM stages (p-value = 1.101e-06) also indicate significant variations, reflecting the disrupted sleep architecture in OSA patients.

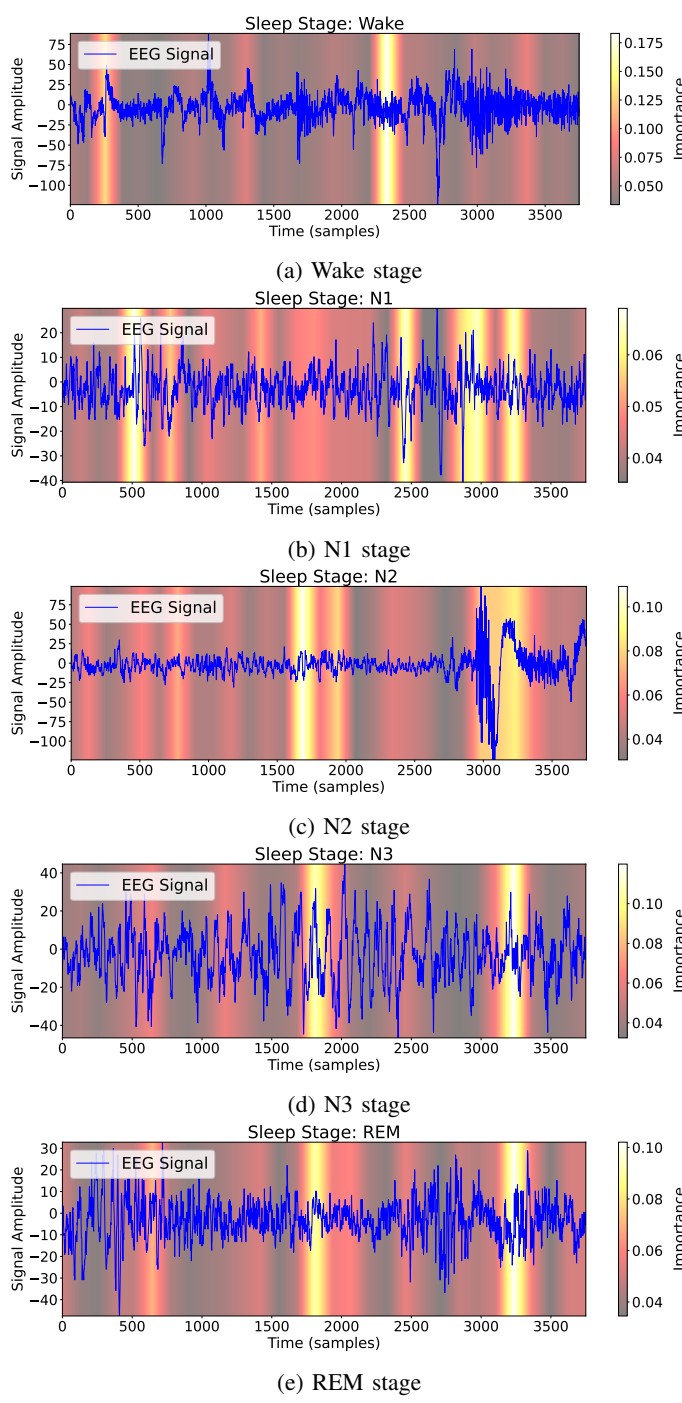

(a) Wake stage

(b) N1 stage

(c) N2 stage

(d) N3 stage

(e) REM stage

Fig. 4: 30 seconds EEG signals with attention heatmap overlay for different sleep stages after 100 epochs training. Regions with higher brightness/heat indicate areas of the signal that have a greater weight during the decision-making process.

## V. CONCLUSION

This study presents a new deep learning framework that enhances the classification of the sleep stage by incorporating self-attention mechanisms and CRF into a model consisting of multi-kernel CNNs and Transformer-based encoders. The self-attention mechanism and CRF module help improve the ability of our model to emphasize discriminative features

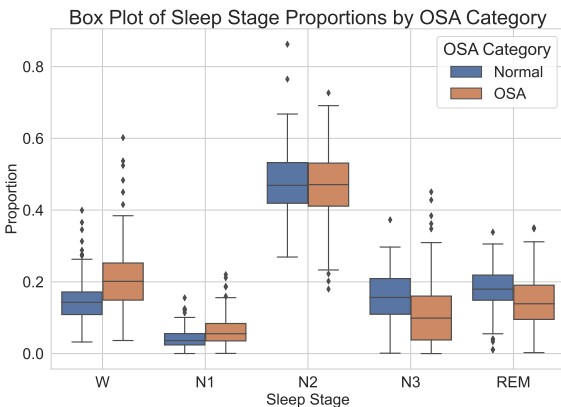

Fig. 5: Box plot showing the proportions of sleep stages (W, N1, N2, N3, REM) by OSA category (Normal and OSA).

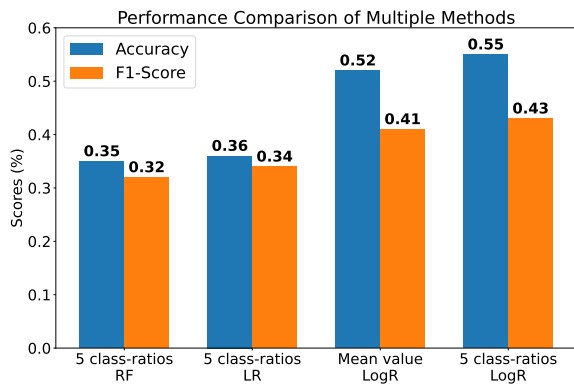

Fig. 6: Performance comparison histogram for five-class classification of OSA severity using different sleep stage features under various regression models.

and capture temporal dependencies between sleep stages, leading to improved classification performance, particularly on datasets with diverse AHI values. Additionally, we have explored the association between sleep stages and the severity of OSA by utilizing the predicted sleep stage features for AHI prediction, revealing the potential of leveraging sleep stage information to assess and monitor sleep apnea. Our model can potentially assist in the early detection and management of sleep disorders, ultimately resulting in improved patient outcomes and enhanced quality of life. Our study demonstrates the promising application of deep learning techniques in uncovering the complex interplay between sleep stages and sleep apnea, laying the foundation for more precise and easily accessible sleep disorder diagnostics.

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
