# OpenReview forum: "Advancing Sleep Disorder Diagnostics: A Transformer-based EEG Model for Sleep Stage Classification and OSA Prediction"
_IEEE.org/EMBS/BHI/2024/Conference — IEEE BHI'24_

### Official Review · Reviewer_VbDN · 2024-08-04
**The paper discusses the importance of precise diagnosis of sleep disorders on health/quality of life. A sequential network model is proposed that improves sleep stage classification by incorporating self-attention mechanisms CRFin a deep learning model composed of CNNs and transformer-based encoders. The self-attention mechanism allows the model to focus on the most discriminative features extracted from single-channel EEG recordings, while the CRF module captures temporal dependencies between sleep stages, improving the model's ability to learn more plausible sleep stage sequences. Furthermore, the paper explores the relationship between sleep stages and OSA severity by using predicted sleep stage characteristics to train various regression models for predicting AHI. The experiments demonstrate an improvement in sleep stage classification of 78.7%, especially on datasets with uniformly distributed AHI values, highlighting the potential of exploiting sleep stage information to monitor OSA. Using advanced deep learning techniques, the study comprehensively explores the complex interplay between sleep stages and sleep apnea, paving the way for more accurate and automated sleep disorder diagnoses. The paper presents various strengths that make it a significant contribution to the field of sleep disorder diagnosis. A strength is the model's ability to capture temporal dependencies between different sleep stages thanks to the integration of the CRF module. This allows the model to learn more plausible and realistic sleep stage sequences, thus increasing classification accuracy. Furthermore, the paper innovatively explores the relationship between sleep stages and the severity of OSA. Using predicted sleep stage characteristics, researchers train various regression models to predict AHI, demonstrating an improvement in sleep stage classification of 78.7%, especially on datasets with uniformly distributed AHI values. This integrated approach not only improves the accuracy of sleep stage diagnosis but also highlights the potential of monitoring OSA through sleep stage analysis. The developed model offers a comprehensive framework for analyzing EEG signals and advancing the diagnosis of sleep disorders, paving the way for more accurate, automated and accessible diagnostic systems. Finally, the ability to provide attention maps to visualize relevant features during the model's decision-making process improves the interpretability and reliability of predictions, making the system also useful for clinical decision support.  However, the paper has some weaknesses. In fact, the proposed model is mainly tested on datasets with uniformly distributed AHI values, which may not reflect the variety and complexity of real clinical data. Model performance may vary in more heterogeneous contexts, where the distribution of the AHI is not uniform. Furthermore, the model relies on single-channel EEG recordings, which, although they simplify the data acquisition process, may not capture all the information necessary for a complete and accurate diagnosis of sleep stages and OSA. This approach may limit the generalizability of the model to more complex clinical cases requiring the analysis of multi-channel EEG signals.  Another weakness concerns the explainability of the model. Although attention maps are used to improve the interpretability of predictions, the inherent complexity of the model, which combines CNNs, transformers, and CRFs, may make it difficult for clinicians to fully understand the system's decision-making process and fully trust its recommendations. Furthermore, model validation was conducted on a limited number of subjects (329), which may not be representative of a larger and more diverse population. Finally, the approach adopted for classifying sleep stages and predicting the severity of OSA does not consider other potentially relevant variables such as age, sex and the presence of other comorbidities, which could significantly influence the results and accuracy of the diagnosis. . For the problem of evaluation on datasets with uniformly distributed AHI values, the experimentation could be expanded using more diversified datasets representative of real clinical conditions, including a greater variety of AHI distributions. This would allow the reliability of the model to be verified in more complex contexts. Regarding the use of single-channel EEG recordings, one possible solution is to integrate multi-channel data into the model, increasing the amount of information available for analysis and thus improving the accuracy of diagnoses. In terms of model explainability, additional visualization and interpretation tools could be developed and incorporated that make the system's decision-making process more understandable to clinicians, fostering greater confidence in the model's recommendations. For validation on a limited number of subjects, the study sample should be expanded to include a larger number of participants and a more diverse population, in order to improve the robustness and generalizability of the results. Finally, to address the lack of consideration of other relevant variables such as age, sex and comorbidities, the model could be enriched with these additional data, allowing for greater personalization of diagnoses and a more complete analysis that takes into account all potentially influential factors. . It should be noted that EEG data require soft computing-based preprocessing due to the fact that they are often affected by uncertainty and/or imprecision.**

**Overall Rating:** 7
**Confidence:** 5

**Other Quality Metrics:**

(a) Clarity of writing: good
(b) Clinical Significance: good
(c) Methodological Novelty: good
(d) Experiments and Results: good

**Questions For The Authors:**

For the problem of evaluation on datasets with uniformly distributed AHI values, the experimentation could be expanded using more diversified datasets representative of real clinical conditions, including a greater variety of AHI distributions. This would allow the reliability of the model to be verified in more complex contexts. Regarding the use of single-channel EEG recordings, one possible solution is to integrate multi-channel data into the model, increasing the amount of information available for analysis and thus improving the accuracy of diagnoses. In terms of model explainability, additional visualization and interpretation tools could be developed and incorporated that make the system's decision-making process more understandable to clinicians, fostering greater confidence in the model's recommendations. For validation on a limited number of subjects, the study sample should be expanded to include a larger number of participants and a more diverse population, in order to improve the robustness and generalizability of the results. Finally, to address the lack of consideration of other relevant variables such as age, sex and comorbidities, the model could be enriched with these additional data, allowing for greater personalization of diagnoses and a more complete analysis that takes into account all potentially influential factors.
It is worth noting that EEG data require preprocessing based on soft computing due to the fact that they are often affected by uncertainty and/or imprecision. So, I ask the authors to insert a sentence in the text that highlights this possibility by listing the following relevant paper in the bibliography:
doi: 10.3390/signals5020018

**Strengths:**

The paper presents various strengths that make it a significant contribution to the field of sleep disorder diagnosis. First, the proposed model combines advanced deep learning techniques, such as multi-kernel convolutional neural networks (CNNs) and transformer-based encoders, with self-attention mechanisms and conditional random fields (CRFs). This combination allows the model to focus on the most relevant features of single-channel electroencephalographic (EEG) recordings, improving the accuracy of sleep stage classification. Another strength is the model's ability to capture temporal dependencies between different sleep stages thanks to the integration of the CRF module. This allows the model to learn more plausible and realistic sleep stage sequences, thus increasing classification accuracy. Furthermore, the paper innovatively explores the relationship between sleep stages and the severity of obstructive sleep apnea (OSA). Using predicted sleep stage characteristics, researchers train various regression models to predict the Apnea-Hypopnea Index (AHI), demonstrating a 78.7% improvement in sleep stage classification, especially on datasets with distributed AHI values. uniformly. This integrated approach not only improves the accuracy of sleep stage diagnosis but also highlights the potential of monitoring OSA through sleep stage analysis. The developed model offers a comprehensive framework for analyzing EEG signals and advancing the diagnosis of sleep disorders, paving the way for more accurate, automated and accessible diagnostic systems. Finally, the ability to provide attention maps to visualize relevant features during the model's decision-making process improves the interpretability and reliability of predictions, making the system also useful for clinical decision support.

**Summary Of The Paper:**

The paper discusses the importance of the precise diagnosis of sleep disorders, in particular obstructive sleep apnea (OSA), on health and quality of life. A sequential network model that improves sleep stage classification is proposed by incorporating self-attention mechanisms and conditional random fields (CRFs) into a deep learning model consisting of multi-kernel convolutional neural networks (CNNs) and transformer-based encoders. The self-attention mechanism allows the model to focus on the most discriminative features extracted from single-channel electroencephalographic (EEG) recordings, while the CRF module captures the temporal dependencies between sleep phases, improving the model's ability to learn sleep phase sequences. more plausible sleep. Furthermore, the paper explores the relationship between sleep stages and OSA severity by using predicted sleep stage characteristics to train various regression models for predicting the Apnea-Hypopnea Index (AHI). The experiments demonstrate an improvement in sleep stage classification of 78.7%, particularly on datasets with uniformly distributed AHI values, highlighting the potential of leveraging sleep stage information to monitor OSA. Using advanced deep learning techniques, the study comprehensively explores the complex interaction between sleep stages and sleep apnea, paving the way for more accurate and automated sleep disorder diagnoses.

**Weaknesses:**

The paper has some weaknesses that deserve attention. One of the main limitations is that the proposed model is mainly tested on datasets with uniformly distributed AHI values, which may not reflect the variety and complexity of real clinical data. Model performance may vary in more heterogeneous contexts, where the distribution of the AHI is not uniform. Furthermore, the model relies on single-channel EEG recordings, which, although they simplify the data acquisition process, may not capture all the information necessary for a complete and accurate diagnosis of sleep stages and OSA. This approach may limit the generalizability of the model to more complex clinical cases requiring the analysis of multi-channel EEG signals. Another weakness concerns the explainability of the model. Although attention maps are used to improve the interpretability of predictions, the inherent complexity of the model, which combines CNNs, transformers, and CRFs, may make it difficult for clinicians to fully understand the system's decision-making process and fully trust its recommendations. Furthermore, model validation was conducted on a limited number of subjects (329), which may not be representative of a larger and more diverse population. Finally, the approach adopted for classifying sleep stages and predicting OSA severity does not consider other potentially relevant variables such as age, sex and the presence of other comorbidities, which could significantly influence the results and the accuracy of diagnosis.

---

### Official Review · Reviewer_eYsg · 2024-08-05
**Review of submission 271**

**Overall Rating:** 7
**Confidence:** 4

**Other Quality Metrics:**

(a)Clarity of writing: 5
(b) clinical significance: 4
(c) Methodological Novelty: 4
(dExperiments and Results: 3

**Questions For The Authors:**

See above.

**Strengths:**

This paper is well organized and well written, presenting a compelling exploration of advanced diagnostic techniques for sleep disorders.
1. The proposed model simultaneously performs sleep stage classification and sleep apnea severity prediction, utilizing multi-kernel CNNs, self-attention mechanisms, and CRFs to adeptly capture temporal dependencies.
2. It effectively handles both short-term sleep stage classification and long-term sleep apnea severity prediction within a single integrated framework.
3. The use of attention heatmaps provides valuable insights into the model’s decision-making process, enhancing both interpretability and reliability.

**Summary Of The Paper:**

Sleep is vital for health, with disorders like obstructive sleep apnea (OSA) causing significant disruptions linked to serious conditions such as hypertension and heart disease. OSA severity is measured by the Apnea-Hypopnea Index (AHI), which categorizes it based on the frequency of breathing interruptions. Additionally, analyzing sleep stages, from light to deep sleep, helps assess sleep quality and health. This paper explores the use of advanced artificial intelligence, specifically deep learning models like Convolutional Neural Networks (CNNs) and Transformer-based models, to analyze EEG signals for sleep disorder diagnostics. While traditional models have focused separately on sleep stage classification or sleep apnea detection, this work proposes a novel approach that combines these tasks. The integration of Transformers, which excel in capturing long-range dependencies and complex patterns, with Conditional Random Fields (CRFs) aims to improve sequence prediction by modeling transitions between sleep stages. This combined model allows for simultaneous classification of sleep stages and prediction of sleep apnea severity, enhancing diagnostic accuracy and efficiency. The study emphasizes model interpretability and introduces attention heatmaps to visualize and understand the decision-making process, contributing to better clinical decision support systems.

**Weaknesses:**

1.There are questions about the model's generalizability, as evidenced by a significant performance drop on a second dataset, indicating sensitivity to data distribution variations.
2. In Table II, the performance of the baseline model attnSleep on N1 is reported as the highest at 30.0, yet the author incorrectly highlights a lower score of 22.0.
3. The improvement attributed to self-attention mechanisms is minimal (only 0.1 improvement on the first dataset), suggesting that self-attention may not significantly enhance performance.
4. The proposed method's performance on N1 classification drops to 6.2 on the second dataset, significantly lower than the baseline model, highlighting a major drawback of the proposed approach.
5. The complexity of the proposed network compared to the baseline network raises questions. It is unclear whether the observed performance gains are due to the proposed architecture or merely the result of increased network complexity.
6. Performance analysis from Table II and Figure 3 does not convincingly demonstrate the effectiveness of self-attention and CRF layers. The authors need to justify why certain classes see significant accuracy improvements without these features and address similar or worse performance metrics like the F1 score and sensitivity.
7. The authors need to provide a clearer explanation of Figure 5, which could enhance understanding of the model's performance nuances.

---

### Official Review · Reviewer_sidU · 2024-08-12
**Advancing Sleep Disorder Diagnostics: A Transformer-based EEG Model for Sleep Stage Classification and OSA Prediction**

**Overall Rating:** 6
**Confidence:** 3

**Other Quality Metrics:**

(a) Clarity of writing; Fair
b) Clinical Significance: Good
(c) Methodological Novelty: Good
(d) Experiments and Results: Good

**Questions For The Authors:**

While the paper is well-written and offers a valuable method for improving sleep stage classification, I have concerns about the generalizability of the results. The exclusion of participants with certain sleep-related disorders may limit the model's applicability to a broader population. Addressing this limitation would strengthen the paper, particularly by considering how other methods in the field tackle this issue and exploring whether similar approaches could be applied here. This would add depth to the analysis and enhance the model's robustness and relevance.

**Strengths:**

The strength of this paper lies in its comprehensive description of the model, which integrates self-attention mechanisms and Conditional Random Fields (CRF) for enhanced sleep stage classification. The study places special emphasis on explainability, effectively capturing and reflecting the distinct neural activities characteristic of each sleep stage. The paper is well-crafted, with its findings clearly and effectively communicated.

**Summary Of The Paper:**

This study proposes an innovative deep learning framework that enhances sleep stage classification by integrating self-attention mechanisms and Conditional Random Fields (CRF) into a model combining multi-kernel Convolutional Neural Networks (CNNs) with Transformer-based encoders. The approach improves the focus on critical features from single-channel EEG recordings and accurately captures temporal dependencies between sleep stages. With a classification accuracy of 78.7%, particularly in datasets with evenly distributed AHI values, this research demonstrates significant potential for monitoring obstructive sleep apnea (OSA). The findings contribute to advancing sleep disorder diagnostics and improving patient outcomes through early detection and management.

**Weaknesses:**

The authors mention that the selection of 329 subjects from a larger pool of 6,441 participants was based on their low Apnea-Hypopnea Index (AHI) values (specifically those with an AHI less than 5). While this approach was intended to minimize interference from severe sleep-related disorders and ensure more stable sleep patterns, it limits the model's exposure to a broader range of sleep disorder severities. As a result, the model's generalizability may be compromised, potentially reducing its effectiveness when applied to populations with higher AHI values or more varied sleep conditions. It would greatly enhance the paper if the authors provided further justification for this selection criteria. Additionally, was the model tested on individuals with an AHI greater than 5 to compare the findings and assess its generalizability? Those results should be included in the manuscript.

---

### Decision · Program_Chairs · 2024-09-23

Accept